# Olive Leaf Extract (OLE) Addition as Tool to Reduce Nitrate and Nitrite in Ripened Sausages

**DOI:** 10.3390/foods11030451

**Published:** 2022-02-03

**Authors:** Graziana Difonzo, Michela Pia Totaro, Francesco Caponio, Antonella Pasqualone, Carmine Summo

**Affiliations:** Department of the Soil, Plant and Food Science DISSPA, University of Bari Aldo Moro Via Amendola, 165/A, I-70126 Bari, Italy; graziana.difonzo@uniba.it (G.D.); michela.totaro@uniba.it (M.P.T.); francesco.caponio@uniba.it (F.C.); antonella.pasqualone@uniba.it (A.P.)

**Keywords:** vegetable extract, by-products, additives, oleuropein, processed meat

## Abstract

Olive leaf extract (OLE) is known to be a source of phenolic compounds with antioxidant and antimicrobial activities. This study investigated the effects of the OLE addition to reduce nitrate/nitrite (NO) content on the physico-chemical features of ripened pork sausages. Seven formulations of pork sausages were set up: CTRL (0 mg/kg OLE; 300 mg/kg NO), Tr1 (200 mg/kg OLE; 150 mg/kg NO), Tr2 (400 mg/kg OLE; 150 mg/kg NO), Tr3 (800 mg/kg OLE; 150 mg/kg NO), Tr4 (200 mg/kg OLE; 0 mg/kg NO), Tr5 (400 mg/kg OLE; 0 mg/kg NO), and Tr6 (800 mg/kg OLE; 0 mg/kg NO). At the end of the ripening period, all the samples were within hygienic limits and the substitution of the additives with OLE allowed the reduction of NO residual contents. Both OLE and NO influenced the colour parameters. At the highest dose of OLE, both alone and in combination with reduced dose of NO, no significant differences in terms of moisture, pH, and a_w_ were found compared to CTRL. In absence of NO, a significant reduction of weight loss was observed. Moreover, in the samples without NO a reduction of the hardness was detected. Finally, the oxidative stability test showed that the increase of the OLE amount prolonged the induction time.

## 1. Introduction

Meat and meat products are a rich source of noble proteins, essential amino acids, vitamins B6 and B12, vitamin D, and various micronutrients [1] and are part of the eno-gastronomic culture of several countries. However, several population studies associated the high meat intake, especially of red and processed meat, with the development of cardiovascular diseases and colorectal cancer in a dose-dependent manner [2]. These risks are mainly due to the high-fat content, particularly richness in saturated fatty acids and cholesterol, and to the carcinogen compounds generated from processing and as a consequence of the additives transformation [3].

Among the latter, nitrites and nitrates are common meat additives which allow: (i) stability of the intense red colour, particularly appreciated by the consumers; (ii) inhibition of the growth of undesirable bacteria, such as *Clostridium botulinum* and toxin production; (iii) improvement of the oxidative stability; (iv) contribution to flavour formation [3,4]. Nevertheless, the addition of nitrates and nitrites to processed meat can induce the development of N-nitroso compounds linked with genotoxicity and metabolic disturbances in the large intestine mucosa [2], with the subsequent risk to potentially develop colorectal cancer [5]. N-nitrosamines (NAs) could also be formed in the human stomach and mouth from residual nitrites in cured-meat products [6,7]. Furthermore, humans are continuously exposed to NAs, considered as environmental pollutants with high toxicity and carcinogenity [3,8]. For these reasons, over the years, great attention has been paid to nitrite reaction products in foods and NAs production [9]. The replacement of nitrate and nitrite by natural extracts in processed meat have been proposed as valuable alternative [10,11].

Over time, a lot of reports about nitrite scavenging and inhibition of NAs formation by natural ingredients, essential oils, extracts from fruits, or vegetables spices have been carried out [3,6]. Furthermore, wastes and by-products from the agricultural and food industry sectors contain highly valuable bioactive substances such as polyphenols, showing antioxidants and antimicrobial activities [12,13]. Polyphenols are known to prevent the initiation or propagation of oxidative reactions [14], decrease lipid and protein oxidation, and inhibit the NAs formation in ripened meat products [3]. Winery waste and by-products (grape seed, grape pomace) have been used in processed meat product formulations to enhance oxidative stability during both ripening and storage periods [15,16]. Other interesting outcomes in the control of protein and lipid oxidation were reported using, respectively, beer residue extract and banana inflorescences [17,18]. Among olive by-products, even the olive leaves have been proposed for food preservation thanks to the abundant bioactive molecules with antioxidant and antimicrobial activities [19,20]. In fact, several studies showed the effects of the addition of olive leaf extract (OLE) to food formulations. OLE improved the oxidative stability [21], exerted an antimicrobial effect [22], and extended the shelf-life of food products [23].

As regards meat products, Aouidi et al. [24] evaluated the effects of OLE on the quality of beef products highlighting the improvement of technological quality, the reduction of defrost and storage loss, and the increase of the oxidative stability of lipid and myoglobin, without adversely affecting the sensory properties. Similarly, Kurt and Ceylan [25] observed a reduction of the oxidation phenomena and bacterial counts in dry-ripened beef sausages added with OLE, further confirming the antimicrobial and antioxidant properties of the extract. However, no studies are available, to the best of our knowledge, on the inclusion of OLE in ripened pork sausages.

In the present work, the feasibility of using OLE for the reduction of nitrate and nitrite level in the ripened sausages was evaluated. To this aim, ripened sausages with different nitrate/nitrite (NO) and OLE ratios were produced in a local farm and characterised in terms of physico-chemical characteristics compared with a control with only NO added.

## 2. Materials and Methods

### 2.1. Production and Characterization of Olive Leaf Extract (OLE)

The production of OLE was carried out as described in Conte et al. [26]. Briefly, after washing, the olive leaves were dried at 120 °C for 8 min in a ventilated oven (Argolab, Carpi, Italy) to reach a moisture content <1%, and then grounded with a blender (Waring-Commercial, Torrington, CT, USA). Milli-Q water was used as extraction solvent in a ratio 1/20 (*w*/*v*). The extraction process was ultrasound-assisted (CEIA, Viciomaggio, Italy) and the extract was filtered through Whatman filter paper (GE Healthcare, Milan, Italy), freeze-dried (BUCHI, Flawil, Switzerland, Lyovapor^TM^ L-200), and stored at −20 °C.

The total phenol content was determined on the extract using the Folin-Ciocalteu method, the antioxidant activity was assessed by ABTS and DPPH assays as reported in Difonzo et al. [19]. The content in oleuropein was determined by HPLC-DAD and external calibration curve with the relative standard, as reported in Centrone et al. [27]. All determinations were performed in triplicate. After the extraction process, the OLE was freeze-dried and stored at −20 °C.

### 2.2. Samples Preparation

Pork meat to produce ripened sausages was purchased from a local farm Salumi Martina Franca S.r.l. (Martina Franca, Italy) where preparation and ripening phases also took place. Sausages were manufactured using the lean meat (shoulder pigs and lean trimmings of whole anatomical piece products) and the adipose tissue (represented by pork belly) (85/15, *w*/*w*) from extensively reared pigs. The raw meats were minced by an industrial meat grinder equipped with a pre-mixer. The grinder meat was separated in seven different batches of 2 kg, each one corresponding to a different combination of NO and OLE concentration (Tr), as reported in Table 1.

The kneading phase was carried out manually to obtain the final homogenous dough and during this step, OLE, nitrate (potassium nitrate E252; SolMar, Taranto, Italy), nitrite (sodium nitrites E250; SolMar, Taranto, Italy), and salt tanning (30% dextrose, 70% salt; 40 g/kg of raw meat) were added. As control sample, a batch without OLE and with the maximum values of additives as defined by Reg. (UE) n. 1129/2011 [28] was prepared. The mixtures were mechanically filled using a natural casing and subjected to stewing (23 °C, RH 95%, 24 h), drying (17–20 °C, RH 60–75%, 96 h), and ripening (15–18 °C, RH 80%).

From each batch, three independent sausage preparations were carried out.

### 2.3. Analytical Determination

#### 2.3.1. Weight Loss, Water Activity (a_w_), pH, Moisture Content

Weight loss (%) was calculated as percentages of differences in weight of the whole sausages between day 0 and the end of ripening time:(1)$$\%\;\mathrm{Weight}\;\mathrm{loss}=\frac{W_{0}-W_{d}\;}{W_{0}}\times \;100$$
where *W*_d_ = weight (g) of the sample at the end of ripening time; *W*_0_ = weight (g) at the production.

The a_w_ was determined using a hygrometer (Aqua Lab 100–240 V AC, Pullman, WA, USA) and the pH was measured with a pH-meter (HANNA instruments, Woonsocket, RI, USA). The determination of moisture content was carried out according to AOAC International methods 950.46 [29]. All the determinations were performed in triplicate on each sausage of different formulations.

#### 2.3.2. Microbiological Analyses

The microbiological parameters considered were mesophilic lactic bacteria count (method ISO 15214:1998) [30], Clostridium sulphite count reducers and spores count (method ISO 15213:2003) [31], coliform count (method ISO 4832:2006) [32], *Escherichia coli* beta-glucuronidase positive count (method ISO 16649-2:2001) [33], coagulase-positive Staphylococci count (method UNI EN ISO 6888-2:2004) [34], and *Listeria monocytogenes* research (method AFNOR BIO 12/27-02/10) [35].

#### 2.3.3. Determination of Residual Nitrate and Nitrite

The residual nitrate content was carried out by spectrophotometric method as reported in Baldini et al. [36]. The nitrite residual content was carried out by the ion-exchange chromatographic method as reported in the UNI EN12014-4:2005 [37].

#### 2.3.4. Colour Measurements

The colour of ripened sausages was analysed according to the International Lighting Commission (CIE) system using the parameters *L** (lightness), *a** (redness), *b** (yellowness) with a colorimeter CM-600d (Konica Minolta, Tokyo, Japan) equipped with the software SpectraMagic NX (Konica Minolta, Tokyo, Japan). The measurements were taken on different points on both surfaces of three 1-cm-thick slices of each sausage [17,38]. The total colour variation was calculated to compare differences between CTRL and ripened sausages with different combinations of OLE/NO:∆E = [(∆*L**)^2^ + (∆*a**)^2^ + (∆*b**)^2^]^1/2^(2)

All the determinations were performed in triplicate on each slice.

#### 2.3.5. Determination of Texture Profile Analysis

Texture Profile Analysis (TPA) was performed with a Texture analyser Z1.0 TN (Zwick Roell, Ulm, Germany) equipped with a cylindrical probe of 36 mm in diameter and a load cell of 1 kN according to Bis-Souza et al. [39] with some modifications. Hardness (N), springiness, gumminess, chewiness (N), and cohesivity (N) values were obtained. After removing the external casing of the sausage, 1-cm-thick portions, having a diameter of 2 cm, were compressed twice up to 50% of recorded deformation at a speed of 1 mm/s. Four different slices of each sausage were considered.

#### 2.3.6. Fatty Acid Determination

The determination of the fatty acid composition of the lipid fraction, extracted using the Folch method [40], was determined by gas chromatography of the fatty acid methyl esters according to the AOCS (American Oil Chemists Society) method Ch 1–91 [41]. A gas-chromatograph 7890B (Agilent Technologies, Palo Alto, CA, USA) was used, equipped with a FID detector and a capillary column SP2340 (60 m × 250 μm × 0.2 μm film thickness) (Supelco Park, Bellefonte, PA, USA). The temperature of the split injector was 210 °C with a splitting ratio of 50:1. The temperature of the column was programmed in a temperature gradient of 80 °C for 5 min, with increments of 2 °C/min, until 200 °C for 5 min, followed by an increase of 10 °C/min, until 240 °C for 5 min. Helium was used as a carrier gas at a constant flow rate of 1 mL/min. The temperature of the FID was set to 220 °C with an air flow of 400 mL/min and helium 40 mL/min. Identification of each fatty acid was carried out by comparing the retention time with that of the corresponding methyl-ester, using a standard mixture (C_4_–C_24_) (Supelco™ 37 component FAME Mix, Bellefonte, PA, USA). The content of individual fatty acids was expressed as relative percentage.

#### 2.3.7. Test of Oxidation Stability

The oxidative stability of sausages was determined with a RapidOxy (Anton Paar, Blankenfelde-Mahlow, Germany) based on forced oxidation of the sample with increase of temperature and O_2_ pressure. The results were expressed in minutes (induction time), which is the time needed for a 10% drop of the oxygen pressure [42]. The set parameters were T, 140 °C, and P, 700 kPa. Sausage samples were minced and tested in triplicate.

### 2.4. Statistical Analysis

Data were subjected to Two-Way ANOVA followed by the Tukey’s HSD test considering the dose of OLE and the dose of NO added as independent variables. Significant differences were determined at *p* < 0.05 by the XLStat software (Addinsoft SARL, New York, NY, USA).

## 3. Results

### 3.1. Characterisation of Olive Leaf Extract (OLE)

Table 2 shows total polyphenolic and oleuropein content, as well as the antioxidant activity of the OLE added to the experimental sausages. These findings are in line with those found in previous works [20,26] in which the authors assayed different properties of OLE in different food matrices and biological systems.

The most abundant phenolic compound was oleuropein, detected by HPLC-DAD, which was quantified in a mean concentration of 93 mg per g of dried extract, as reported in Table 2.

### 3.2. Microbiological Analysis, Nitrate and Nitrite Residual

Microbiological analysis of the ripened sausages represents a crucial point for hygiene and health safety of these products [43,44] and the addition of the NO is a technological action to prevent the growth of undesirable microorganisms that can affect the hygiene and the quality of the products.

In the samples investigated, all the microbiological parameters evaluated (sulphite-reducing *Clostridia* and spores, Coliforms, *Escherichia coli* and *Staphylococcus* coagulase positive) were within the law limits Reg. (CE) n. 1441/2007 [45] (data not shown). A similar trend was observed in other studies focused on the substitution of conventional additives with natural extracts [15,46] and powder [47,48,49]. Aquilani et al. [15] showed that all major foodborne pathogens were absent or below the limit required in dry-fermented sausages manufactured with grape seed extract and chestnut extract as the replacement of sodium nitrite. Additionally, Pateiro et al. [46] obtained a lower total viable count in dry-cured sausages with natural extracts (tea, chestnut, grape seed, and beer extracts). Ozaki et al. [47] demonstrated that after 35 days of drying and after 60 days of storage, all dry sausages with beetroot and radish powder as natural nitrite could be consumed safely and no thermotolerant coliform bacteria were found in all treatments. Other works assayed the antimicrobial potential of OLE in different food products, such as olive preserves and fresh dairy products [21,50,51]. The inclusion of OLE in the olive-based pâté decreased the microbial number in treated samples and no pathogen growth was observed [50]. Additionally, the combination of OLE with commercial starter in table olives [21] and synergic effect of OLE with ascorbic acid [51] may protect table olive spoilage and pathogens during the fermentation process [21] and *Enterococcus* spp. and *Enterobacteriaceae* in Stracciatella cheese, an Italian specialty [51].

Therefore, a possible role of OLE in exerting an antimicrobial effect can be assumed since also in the samples without NO added no growth was found. This could be due to the direct inhibitory action of OLE and the presence of polyphenolic compounds in OLE with antimicrobial properties [21,50]. The specific antimicrobial mechanism of polyphenols in OLE is unknown [52]. However, the ability to reduce the growth rate of microorganisms, inhibit several enzymes, micrococcal nuclease, lysozyme, and cause damage to cell membranes has been ascertained [52,53]. Liu et al. [53] showed that OLE is effective in inhibiting the growth of some foodborne pathogens, reducing their mobility. At the same time, the oleuropein, the most abundant polyphenolic compound of OLE [19,54], and verbascoside have a great impact against some foodborne pathogens [53]. The oleuropein could inhibit microbial growth by determining changes in gene expression [53,55]. Furthermore, it was also noted that verbascoside could be more effective than oleuropein in the inhibition of bacterial growth; in fact, at the same concentration, verbascoside was more effective in inhibiting some pathogens [53].

In Table 3 the residual nitrate and nitrite content of the samples under investigation are reported.

The highest residual nitrate values (15 mg/kg) were found in the CTRL sample (OLE_0_, NO_300_). In the samples without NO, amounts of residual nitrate below 5 mg/kg were found. Similar results were found for nitrite determination, which were higher in CTRL (20 mg/kg) and were reduced in all other samples with an estimated maximum content of 11 mg/kg. It can be also observed that the residual nitrite content was higher than the nitrate, probably due to the conversion of nitrate to nitrite by nitrate-reductase bacteria [47,56].

Our findings agree with those obtained in other studies. Martínez-Zamora et al. [56] found that residual nitrate and nitrite content of Spanish chorizo with leafy green vegetables as natural nitrate sources were significantly higher in a control sample made with a commercial mix with synthetic nitrate. Moreover, the residual nitrite content of control sample of Turkish fermented beef sausage was higher in comparison with those of the samples including beetroot powder [57]. Other authors showed that the addition of 1% radish powder turned out be the best option for NO contents in fermented dry sausages [47]. In another study with cooked sausages, the highest residual nitrite content was found in the sample added with 150 mg/kg sodium nitrite, whereas the lowest was observed in treatments with pre-converted nitrites from natural sources (spinach, lettuce, celery, and red beet) [58].

It is well known that NO is involved in process stability of ripened sausages [47]. In our study, the NO residual content seemed to be strictly correlated with the NO dose added, although the EFSA [59] reported that other factors, such as processing conditions, pH, and presence of reductants, could be more relevant.

Our results highlighted that the substitution of NO with the OLE allowed the reduction of the residual contents, whose presence in meat products is highly questionable based on the evidence of the relationship between N-nitrosamines and the increase of the risk of cancer [5]. In fact, it was observed that the high reactivity of nitrite enables it to react with bioactive compounds; thus, reducing the residual nitrite level and its reaction with the polyphenols compound could lead to less formation of N-nitroso compounds [60].

### 3.3. Qualitative Characterization of Ripened Sausages

The main qualitative features such as the physical characteristics, pH, a_w_, and moisture (%) of the ripened sausages are shown in Table 4.

Both the variables under investigation (dose of OLE and dose of NO added) influenced all the qualitative features considered (*p* < 0.05).

Considering moisture, the highest value was found for Tr5, which was also significantly higher than CTRL, whereas no significant differences were found among the other samples. These outcomes not are totally in line with data reported by other authors who observed lower moisture contents in all sausages prepared with vegetables compared with the control [47,61]. Considering NO dose added, the trend observed is, however, clear. The reduction of the 50% of the NO did not cause a variation of the water content, while the absence of NO determined an increase of the water content. These results could be linked to the absence of osmotic dehydration induced by the NO [62]. In fact, it was highlighted how the nitrite penetration in the samples takes place simultaneously with the decrease in water-holding capacity, leading to a reduction in water content [62].

The values of weight loss did not show statistical differences in the samples with NO_150_ (Tr1, Tr2, Tr3) versus CTRL; whereas, for the samples added only with the OLE (Tr4, Tr5, Tr6), values statistically lower than CTRL were observed. Considering the samples with different OLE concentrations, a dose-dependent increase in weight loss trend could be observed.

It can be considered that the lower moisture values observed in ripened sausage with OLE could be explained by the content of fibres of OLE [63], which could facilitate the water loss, also determining the higher weight loss (Tr1, Tr2, Tr3, and Tr6) [47]. In other studies, only the inclusion of grape seed and chestnut extracts [15], functional ingredients from olive and citrus extracts [64] and spices, fruits, and vegetables [56] in the formulation of dry-fermented sausages, Spanish chorizo, and Spanish-Type dry-cured sausage “Fuet”, respectively, did not affect the weight loss compared to the control with sodium nitrite.

As expected, the trend of the moisture and weight loss were strictly correlated with the pH evolution. When pH went closer to the isoelectric point, it induced meat proteins to partially leak their water retention capacity, causing a subsequent water loss [65].

The reduction of the 50% of the added NO dose did not cause a significant variation in pH value, while, when the additives were not added, the trend increased. This outcome could be related to the production of ammonia and other basic substances caused by protein breakdown resulting from the growth of microorganisms [38,66] or to the buffering capacity of meat [57,67]. However, the increase of pH in sample without NO was mitigated by the inclusion of the highest dose of OLE added (800 mg/kg). This trend might be linked to the higher lactic acid production by lactic bacteria during ripening [65], due to the higher availability of fermentable carbohydrates from OLE. In fact, it was noted that the carbohydrate composition of OLE extract might have provided additional substrate for microorganisms [63,67]. In this regard, Aquilani et al. [15] found a lower value of pH in a sample of dry-fermented sausages with grape seed extract and chestnut extract than control with sodium nitrite, suggesting that lactic bacteria growth could be slightly promoted in these products [15] with natural NO substitutes.

Our outcome agrees with data reported by Ozaki et al. [47], who observed an increase in the pH values for dry-ripened sausages with beetroot and radish powder. On the contrary, Kurt and Ceylan [25] found no changes in pH by the addition of the OLE in dry-fermented beef sausages at the end of the ripening period.

As regards the a_w_ trend, on the whole, all samples presented values lower than 0.89, positively contributing to control the development of pathogen microorganisms [15]. Specifically, the highest values were found in Tr4 and Tr5 (a_w_ = 0.81), since the addition of OLE could represent a barrier to the water loss, probably because of its water holding, as it was shown for radish powder in fermented cooked sausages [49], even if Tr6 did not show the same trend. This different trend it could be due to any different ripening conditions. However, our trend appears in line with a study of Aquilani et al. [15], in which no significant differences in terms of a_w_ were found between the dry-fermented sausages with grape seed and chestnut extract and those with sodium nitrite.

### 3.4. Colour Analysis

The addition of vegetable extracts in meat products can affect their colour features [18]. Any changes of this parameter are linked to consumer acceptability [68]. Table 5 shows the results of colour determination of ripened sausages.

Colour parameters were found to be affected by the doses of NO and OLE added (Figure 1). Lightness (*L**), redness (*a**), and yellowness (*b**) were significantly linked to the addition of NO (*p* < 0.001, *p* < 0.001 and *p* = 0.025, respectively). Furthermore, yellowness (*b**) was also influenced by the OLE variable (*p* < 0.001).

The absence of NO induced a significant reduction of the *a** index, while the different OLE doses added did not significantly affect this important parameter. Redness of meat products is one of the main factors considered by the consumers, influencing the willingness to purchase. Increasing appreciation for bright hues was reported, being a bright hue perceived by consumers as an indicator of quality [69,70]. The lowest *a** value was observed in NO_0_ added sausages, probably linked to the lack of synthesis of nitroso-myoglobin, which determines the red colour stability of the meat based product, due in turn to the missing reaction between nitric oxide and myoglobin in the samples [68]. Our trend agrees with other studies [18,25], which observed a reduction of *a** with the addition of 1.5 and 2% of banana inflorescence [18] and olive leaf extract [25] in ripened beef sausages formulation.

It could be observed that redness values found in our study were lower than that of ripened sausages of the other studies [18,71] due to the raw meat chosen for this research, represented by the *Apulo Calabrese black pig* breed, which is characterized by a naturally darker shade of red meat colour.

The reduction of the NO doses caused a significant increase of the lightness, while no significant effect was linked to the OLE variables. The lightness of the sausages could be related to the different moisture content of the samples. The higher moisture content, as the values observed in our samples at reduced or absent NO, caused an increase of the wet surface and a consequence increase of the light scatter [72]. The OLE variable significantly affected the yellowness index (*b**). A reduction of the parameter was observed to increase the OLE doses. Other studies showed the same trend [18,64]. Particularly, yellowness has great influence on meat discoloration [73] generally associated to the accumulation of metmyoglobin on the meat surface [74]. Wang et al. [73] showed that yellowness is positively correlated with different factors, such as microbial spoilage, myoglobin autoxidation, lipid oxidation, and protein oxidation. Thus, among the phenomena that determine the formation of yellow pigments, there are factors associated with oxidation processes, and it is reasonable to consider that the presence of antioxidant compounds of OLE could have acted as efficient radical scavengers and metal chelators, reducing oxidative processes which promote colour changes [74].

In addition, ΔE was also determined to compare the evolution of visual colour between control and other treatments. ΔE values were between 2.23 and 7.73, meaning that the observer can perceive the difference between samples [1], since values higher than 2.0 indicate “visually perceptible” differences [47]. A decrease of this parameter was observed in the trials with NO_150_. Particularly, the changes occurring in Tr2 and Tr3 can even observed by an unexperienced observer, since 2.0 < ∆E < 3.5 [1,75].

### 3.5. Texture Profile Analysis

Considering the hardness, both OLE and NO variables significantly influenced this parameter (*p* < 0.001), and except for Tr2 and Tr3, all samples showed significantly lower value than CTRL (108 N). The reduction of the hardness was noticeable in absence of NO (NO_0_) and less marked in the intermedial NO doses (NO_150_) (Table 6).

Our outcome agrees with a study of Martínez-Zamora [56], in which lower values of the hardness were found in sample of fermented sausages added with spices, fruits, and vegetables [56] than the sample control. The authors linked this trend to the presence of synthetic NO source and thiol loss due to proteolysis produced by protein oxidation and the natural ripening process [56]. Furthermore, in our study, although without significant differences, in both sample groups with NO_150_ and NO_0_, the hardness increased by increasing the dose of OLE. This could be due to the increase in weight loss of sample in each group. Nevertheless, Tsoukalas et al. [76] did not find differences in hardness of fermented sausages by adding freeze-dried leek powder at different doses of 0.84% and 1.68%.

As for the other TPA parameters evaluated (springiness, gumminess, chewiness, and cohesivity), except for Tr6 gumminess and chewiness values, we observed a significant increase in the samples OLE_800_/NO_150_ and OLE_800_/NO_0_ compared with the control. Our trend is in line with that found in a study of Aquilani et al. [15]; in fact, cohesivity and springiness of the sausages elaborated with chestnut extract were higher than the control sample elaborated with sodium nitrite. This could be attributed to the content of polysaccharides [77] of OLE, which could have gel-forming capability and could help to form aggregates, as it was shown with other natural nitrite alternative in sausages [10]. The springiness, cohesivity, and chewiness are strictly related to the pH of the sample, and also slight variations could affect the parameters [15]. Higher values of pH could inhibit the aggregation of myofibrillar proteins and leads to gel formation [15]. This fact could be also demonstrated comparing Tr4 and Tr5 to Tr6 with only OLE added. Indeed, Tr4 and Tr5 presented higher values of pH than Tr6, resulting in the reduction of cohesivity and chewiness, probably due to the correlation between the presence of OLE and pH, as we already discussed in Section 3.3.

### 3.6. Fatty Acids Composition

The main fatty acids found in the ripened sausages are reported in Table 7.

The main fatty acids percentage at the end of ripened period in all samples showed a composition like MUFA > SFA > PUFA as previously observed by Pini et al. [11]. Oleic acid, palmitic acid, stearic acid, myristic acid, and linolenic acid were mainly found. Interestingly, the C_18:3_ relative content found in our samples was higher than those reported in other meat and meat product from pigs reared extensively [78,79]. Omega3 fatty acids, such as linolenic acid, are essential fatty acids that, if present in appropriate content, are considered useful in preventing coronary heart disease and decreasing blood clotting [80]. In general, slight changes were detected for the lipid fractions comparing sausages with different combinations of OLE/NO and control. Particularly, for PUFA content, there were no statistical differences, even if sample with NO_150_ showed the lowest value. Pini et al. [11] observed changes in unsaturated fatty acids fractions when comparing fermented sausages with natural extracts and control with sodium nitrite. The inclusion of grape seed extract in the samples showed a lower value of PUFA [11]. It can be considered that differences in the fatty acids composition of the ripened sausages could be linked to a different evolution of the oxidative phenomena occurring during ripening. Since PUFA double bonds as preferred substrates for oxidative reactions [11,15,46], our results suggested a lower oxidation of this lipid fraction in OLE samples than the control and other samples with OLE/NO, probably be due to the antioxidant activity of the OLE, as previously demonstrated [19,20,21,51].

### 3.7. Test of Oxidation Stability

The oxidative stability test was carried out by RapidOxy and showed a trend reported in Figure 2. The antioxidant efficacy of OLE was already observed in raw and cooked meat but never in ripened meat [20]. Polyphenols inhibit the oxidative processes in three ways: as reactive species scavengers, lipoxygenase inhibitors, and reducing agents for metmyoglobin. Further, NO can exert antioxidant activity by inhibiting lipid oxidation and chelating metals [81].

The samples with the maximum amount of the OLE (800 mg/kg) showed the higher induction time (Tr3 = 73.68 min; Tr6 = 63.37 min) and thus the highest oxidative stability. Tr3 showed the highest value, and it was also significantly higher compared with CTRL. On the contrary, Tr6 showed a value significantly higher only compared with Tr4. The trend could be confirmed with what we also suggested before. In fact, the increase of the OLE amount corresponded to a greater induction time. However, the synergy with NO cannot be excluded, since the sample with the lowest content of OLE and no added NO (Tr4) showed the lowest oxidative stability, while the homolog with NO added (Tr1) showed higher values of induction time.

Overall, the results indicated that OLE, both alone and together with NO, exerted an antioxidant effect in ripened sausages and, interestingly, the OLE addition bolstered the antioxidant power of NO.

## 4. Conclusions

Our research highlighted the effectiveness of OLE extract in reducing NO in the formulation of ripened pork sausages. Together with an effective reduction of the NO residual, the replacement did not affect the hygiene and safety at the end products. The highest OLE dose, both alone and in combination with the reduced dose of NO, did not cause any significant differences in terms of moisture content, pH, and a_w_ compared to CTRL. In absence of NO, a significant reduction of weight loss was observed. Colorimetric parameters were significantly linked to the addition of NO and yellowness was also influenced by the OLE variable. Except for Tr2 and Tr3, the inclusion of OLE caused a significant decrease in hardness compared with CTRL. Overall, in the samples without NO, the OLE at the highest concentration (800 mg/kg) mitigated the pH increasement and positively influenced some textural parameters, such as springiness and cohesivity.

The results of the main fatty acids composition and oxidation stability test proved the antioxidant effect of the extract, mostly at the concentration of 800 mg/kg.

The interesting results observed in our research represent a step towards the increase of meat products safety while improving their sustainability.

## Figures and Tables

**Figure 1 foods-11-00451-f001:**
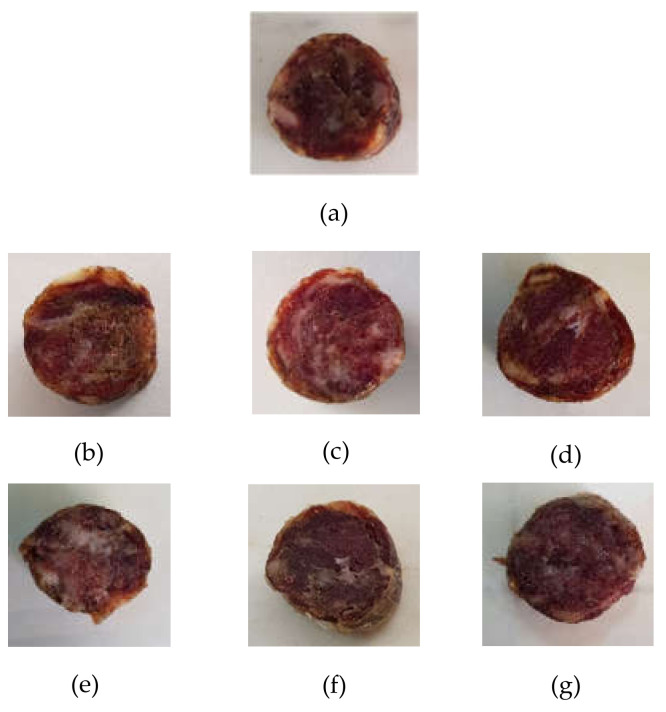
Colour variation of the ripened sausages under investigation: (**a**) CTRL (0 mg/kg OLE; 300 mg/kg NO); (**b**) Tr1 (200 mg/kg OLE; 150 mg/kg NO); (**c**) Tr2 (400 mg/kg OLE; 150 mg/kg NO); (**d**) Tr3 (800 mg/kg OLE; 150 mg/kg NO); (**e**) Tr4 (200 mg/kg OLE; 0 mg/kg NO); (**f**) Tr5 (400 mg/kg OLE; 0 mg/kg NO); (**g**) Tr6 (800 mg/kg OLE; 0 mg/kg NO).

**Figure 2 foods-11-00451-f002:**
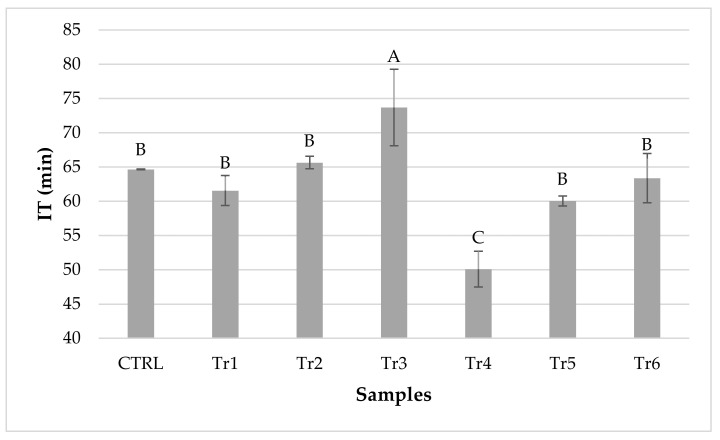
Results of the RapidOxy Test of the ripened sausages. CTRL, control; Tr, trial; IT, Induction Time. CTRL (0 mg/kg OLE; 300 mg/kg NO), Tr1 (200 mg/kg OLE; 150 mg/kg NO), Tr2 (400 mg/kg OLE; 150 mg/kg NO), Tr3 (800 mg/kg OLE; 150 mg/kg NO), Tr4 (200 mg/kg OLE; 0 mg/kg NO), Tr5 (400 mg/kg OLE; 0 mg/kg NO), Tr6 (800 mg/kg OLE; 0 mg/kg NO). Different letters indicate significant differences at *p* < 0.05.

**Table 1 foods-11-00451-t001:** Formulations of OLE—NO used for samples preparation.

	OLE (mg/kg)	NO (mg/kg)
CTRL	0	150 NO_2_/150 NO_3_
Tr1	200	75 NO_2_/75 NO_3_
Tr2	400	75 NO_2_/75 NO_3_
Tr3	800	75 NO_2_/75 NO_3_
Tr4	200	0
Tr5	400	0
Tr6	800	0

OLE, Olive Leaf Extract; NO, Nitrate/nitrite; CTRL, control; Tr, trial; CTRL (0 mg/kg OLE; 300 mg/kg NO), Tr1 (200 mg/kg OLE; 150 mg/kg NO), Tr2 (400 mg/kg OLE; 150 mg/kg NO), Tr3 (800 mg/kg OLE; 150 mg/kg NO), Tr4 (200 mg/kg OLE; 0 mg/kg NO), Tr5 (400 mg/kg OLE; 0 mg/kg NO), Tr6 (800 mg/kg OLE; 0 mg/kg NO).

**Table 2 foods-11-00451-t002:** Total polyphenolic content (TPC), antioxidant activity (ABTS, DPPH) and oleuropein content of the olive leaf extract (*n* = 3).

Parameter	Amount
TPC (mg GAE/g)	126.35 ± 1.39
ABTS (µmol TE/g)	709.17 ± 8.57
DPPH (µmol TE/g)	676.59 ± 3.25
Oleuropein (mg/g)	93.12 ± 2.21

GAE, Gallic acid equivalent; TE, Trolox equivalents.

**Table 3 foods-11-00451-t003:** Residual nitrate and nitrite content (mg/kg).

	OLE (mg/kg)	NO (mg/kg)	Residual Nitrate (mg/kg)	Residual Nitrite (mg/kg)
CTRL	0	300	15	20
Tr1	200	150	6	<10
Tr2	400	150	10	<10
Tr3	800	150	9	11
Tr4	200	0	<5	<10
Tr5	400	0	<5	<10
Tr6	800	0	<5	<10

OLE, Olive Leaf Extract; NO, Nitrate/nitrite; CTRL, control; Tr, trial; CTRL (0 mg/kg OLE; 300 mg/kg NO), Tr1 (200 mg/kg OLE; 150 mg/kg NO), Tr2 (400 mg/kg OLE; 150 mg/kg NO), Tr3 (800 mg/kg OLE; 150 mg/kg NO), Tr4 (200 mg/kg OLE; 0 mg/kg NO), Tr5 (400 mg/kg OLE; 0 mg/kg NO), Tr6 (800 mg/kg OLE; 0 mg/kg NO).

**Table 4 foods-11-00451-t004:** Mean value, standard deviation, and results of the statistical analysis (two-way ANOVA) of moisture, weight loss, pH, and water activity (a_w_) of the ripened sausages under investigation (*n* = 3).

	CTRL	Tr1	Tr2	Tr3	Tr4	Tr5	Tr6	*p*-Value
Moisture (%)	22.37 ± 0.18 ^BC^	20.25 ± 0.91 ^C^	22.08 ± 0.76 ^BC^	21.75 ± 0.82 ^BC^	23.67 ± 1.08 ^AB^	25.27 ± 0.90 ^A^	20.54 ± 0.31 ^C^	OLE = 0.025 NO = 0.019
Weight loss (%)	44.81 ± 0.63 ^A^	45.37 ± 0.21 ^A^	44.70 ± 1.37 ^A^	45.60 ± 0.55 ^A^	39.47 ± 0.63 ^B^	38.47 ± 1.87 ^B^	41.19 ± 0.94 ^B^	OLE = 0.041 NO < 0.001
pH	5.24 ± 0.00 ^C^	5.12 ± 0.00 ^D^	5.18 ± 0.03 ^C^	5.20 ± 0.05 ^C^	5.30 ± 0.01 ^B^	5.43 ± 0.00 ^A^	5.22 ± 0.00 ^C^	OLE = 0.021 NO < 0.001
a_w_	0.75 ± 0.00 ^B^	0.75 ± 0.00 ^B^	0.75 ± 0.00 ^B^	0.75 ± 0.00 ^B^	0.81 ± 0.00 ^A^	0.81 ± 0.00 ^A^	0.75 ± 0.00 ^B^	OLE = 0.013 NO < 0.001

OLE, Olive Leaf Extract; NO, Nitrate/nitrite; CTRL, control; Tr, trial; CTRL (0 mg/kg OLE; 300 mg/kg NO), Tr1 (200 mg/kg OLE; 150 mg/kg NO), Tr2 (400 mg/kg OLE; 150 mg/kg NO), Tr3 (800 mg/kg OLE; 150 mg/kg NO), Tr4 (200 mg/kg OLE; 0 mg/kg NO), Tr5 (400 mg/kg OLE; 0 mg/kg NO), Tr6 (800 mg/kg OLE; 0 mg/kg NO). Different letters indicate significant differences at *p* < 0.05.

**Table 5 foods-11-00451-t005:** Mean value, standard deviation, and results of the statistical analysis (two-way ANOVA) of the colour parameters of the ripened sausages under investigation (*n* = 3).

	CTRL	Tr1	Tr2	Tr3	Tr4	Tr5	Tr6	*p*-Value
**Lightness (*L**)**	32.58 ± 0.02 ^B^	35.37 ± 2.92 ^AB^	33.98 ± 0.39 ^B^	33.25 ± 0.74 ^B^	35.18 ± 0.62 ^AB^	37.92 ± 0.58 ^A^	35.43 ± 0.75 ^AB^	OLE = 0.213 NO < 0.001
**Redness (*a**)**	8.27 ± 0.60 ^A^	8.48 ± 1.58 ^A^	8.10 ± 1.72 ^A^	6.55 ± 1.27 ^AB^	4.24 ± 0.19 ^B^	3.35 ± 1.24 ^B^	4.42 ± 1.06 ^B^	OLE = 0.928 NO < 0.001
**Yellowness (*b**)**	7.55 ± 0.18 ^AB^	9.27 ± 1.33 ^AB^	6.85 ± 0.30 ^B^	6.57 ± 0.30 ^B^	8.26 ± 0.74 ^AB^	9.90 ± 2.13 ^A^	6.92 ± 0.03 ^B^	OLE < 0.001 NO = 0.025
**ΔE vs. _CTRL_**		3.63 ± 2.97	2.30 ± 0.60	2.23 ± 0.51	4.94 ± 0.55	7.73 ± 2.41	4.89 ± 0.06	

OLE, Olive Leaf Extract; NO, Nitrate/nitrite; CTRL, control; Tr, trial; CTRL (0 mg/kg OLE; 300 mg/kg NO), Tr1 (200 mg/kg OLE; 150 mg/kg NO), Tr2 (400 mg/kg OLE; 150 mg/kg NO), Tr3 (800 mg/kg OLE; 150 mg/kg NO), Tr4 (200 mg/kg OLE; 0 mg/kg NO), Tr5 (400 mg/kg OLE; 0 mg/kg NO), Tr6 (800 mg/kg OLE; 0 mg/kg NO). Different letters indicate significant differences at *p* < 0.05.

**Table 6 foods-11-00451-t006:** Mean value, standard deviation, and results of the statistical analysis (two-way ANOVA) of the textural parameters of the ripened sausages under investigation (*n* = 3).

	CTRL	Tr1	Tr2	Tr3	Tr4	Tr5	Tr6	*p*-Value
Hardness (N)	108.00 ± 0.00 ^A^	74.13 ± 12.83 ^BC^	84.00 ± 13.44 ^AB^	97.18 ± 5.45 ^AB^	43.87 ± 1.95 ^C^	69.78 ± 22.42 ^BC^	71.36 ± 3.01 ^BC^	OLE = 0.007 NO < 0.001
Springiness	0.37 ± 0.04 ^C^	0.43 ± 0.02 ^AB^	0.42 ± 0.02 ^ABC^	0.47 ± 0.00 ^A^	0.39 ± 0.03 ^BC^	0.37 ± 0.01 ^BC^	0.46 ± 0.00 ^A^	OLE < 0.001 NO < 0.001
Gumminess	18.64 ± 2.48 ^B^	16.05 ± 3.89 ^BC^	17.62 ± 2.35 ^B^	29.68 ± 1.38 ^A^	10.84 ± 0.23 C	15.68 ± 3.02 ^BC^	20.95 ± 0.39 ^B^	OLE < 0.001 NO < 0.001
Chewiness (N)	7.07 ± 1.65 ^BCD^	7.13 ± 2.11 ^BCD^	7.61 ± 0.70 ^BC^	14.69 ± 0.41 ^A^	4.28 ± 0.41 ^D^	5.94 ± 1.11 ^CD^	9.75 ± 0.15 ^B^	OLE < 0.001 NO < 0.001
Cohesivity (N)	0.17 ± 0.03 ^D^	0.21 ± 0.01 ^CD^	0.20 ± 0.01 ^CD^	0.29 ± 0.03 ^A^	0.25 ± 0.01 ^ABC^	0.23 ± 0.03 ^BC^	0.28 ± 0.01 ^AB^	OLE < 0.001 NO = 0.135

OLE, Olive Leaf Extract; NO, Nitrate/nitrite; CTRL, control; Tr, trial; CTRL (0 mg/kg OLE; 300 mg/kg NO), Tr1 (200 mg/kg OLE; 150 mg/kg NO), Tr2 (400 mg/kg OLE; 150 mg/kg NO), Tr3 (800 mg/kg OLE; 150 mg/kg NO), Tr4 (200 mg/kg OLE; 0 mg/kg NO), Tr5 (400 mg/kg OLE; 0 mg/kg NO), Tr6 (800 mg/kg OLE; 0 mg/kg NO). Different letters indicate significant differences at *p* < 0.05.

**Table 7 foods-11-00451-t007:** Mean value, standard deviation, and results of statistical analysis of the main fatty acids of the lipid fraction of the ripened sausages (*n* = 3).

	CTRL	Tr1	Tr2	Tr3	Tr4	Tr5	Tr6	*p*-Value
Myristic C14:0	1.34 ± 0.01 ^AB^	1.41 ± 0.05 ^A^	1.33 ± 0.02 ^AB^	1.33 ± 0.05 ^AB^	1.32 ± 0.02 ^B^	1.34 ± 0.01 ^AB^	1.34 ± 0.01 ^AB^	OLE = 0.305 NO = 0.178
Palmitic C16:0	23.32 ± 0.03 ^B^	24.41 ± 0.64 ^A^	23.79 ± 0.17 ^AB^	23.65 ± 0.51 ^AB^	23.36 ± 0.23 ^B^	23.83 ± 0.08 ^AB^	23.55 ± 0.08 ^AB^	OLE = 0.513 NO = 0.085
Stearic C18:0	10.80 ± 0.18 ^B^	11.76 ± 0.10 ^A^	11.50 ± 0.09 ^A^	11.55 ± 0.03 ^A^	11.19 ± 0.55 ^AB^	11.73 ± 0.07 ^A^	11.22 ± 0.06 ^AB^	OLE = 0.420 NO = 0.122
Oleic C18:1	48.44 ± 0.22 ^A^	47.54 ± 0.13 ^AB^	48.48 ± 0.19 ^A^	48.20 ± 0.39 ^A^	47.74 ± 0.55 ^A^	46.77 ± 0.15 ^B^	47.82 ± 0.47 ^A^	OLE = 0.425 NO = 0.033
Linoleic C18:2	9.95 ± 0.08 ^A^	9.04 ± 0.73 ^A^	9.07 ± 0.09 ^A^	9.21 ± 0.99 ^A^	10.27 ± 0.15 ^A^	10.25 ± 0.12 ^A^	10.03 ± 0.44 ^A^	OLE = 0.787 NO < 0.001
Linolenic C18:3	1.10 ± 0.01 ^AB^	1.07 ± 0.02 ^B^	1.08 ± 0.02 ^B^	1.12 ± 0.00 ^A^	1.07 ± 0.01 ^B^	1.02 ± 0.01 ^C^	1.07 ± 0.02 ^B^	OLE = 0.008 NO = 0.002
ƩSFA	35.93 ± 0.13 ^B^	38.08 ± 0.86 ^A^	37.07 ± 0.30 ^AB^	37.12 ± 0.68 ^AB^	36.32 ± 0.76 ^B^	37.35 ± 0.02 ^AB^	36.56 ± 0.01 ^B^	OLE = 0.595 NO = 0.059
ƩMUFA	52.03 ± 0.25 ^A^	50.95 ± 0.09 ^BC^	51.93 ± 0.20 ^AB^	51.62 ± 0.40 ^AB^	51.33 ± 0.60 ^ABC^	50.37 ± 0.14 ^C^	51.39 ± 0.48 ^AB^	OLE = 0.467 NO = 0.103
ƩPUFA	12.02 ± 0.12 ^A^	10.96 ± 0.77 ^A^	10.99 ± 0.09 ^A^	11.24 ± 1.08 ^A^	12.34 ± 0.16 ^A^	12.27 ± 0.12 ^A^	12.04 ± 0.48 ^A^	OLE = 0.719 NO < 0.001
n-3 PUFA	1.19 ± 0.02 ^AB^	1.14 ± 0.01 ^CD^	1.1 ± 0.01 ^CD^	1.20 ± 0.02 ^A^	1.18 ± 0.01 ^ABC^	1.13 ± 0.00 ^D^	1.16 ± 0.01 ^BCD^	OLE = 0.032 NO = 0.393

SFA, Saturated fatty acids; MUFA, Monounsaturated fatty acids; PUFA, Polyunsaturated fatty acids; OLE, Olive Leaf Extract; NO = Nitrate/nitrite; CTRL, control; Tr, trial; CTRL (0 mg/kg OLE; 300 mg/kg NO), Tr1 (200 mg/kg OLE; 150 mg/kg NO), Tr2 (400 mg/kg OLE; 150 mg/kg NO), Tr3 (800 mg/kg OLE; 150 mg/kg NO), Tr4 (200 mg/kg OLE; 0 mg/kg NO), Tr5 (400 mg/kg OLE; 0 mg/kg NO), Tr6 (800 mg/kg OLE; 0 mg/kg NO). Different letters indicate significant differences at *p* < 0.05.

## Data Availability

The data presented in this study are available on request from the corresponding author.

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
