# Peer review of "Olive Leaf Extract (OLE) Addition as Tool to Reduce Nitrate and Nitrite in Ripened Sausages"

_foods, 2022, doi:10.3390/foods11030451_

Round 1

Reviewer 1 Report

In general, the manuscript is well written and organized. However, some apsects must be addressed to improve it:

  • Check the English grammar in the whole manuscript. For example in the title, is a tool the OLE by itself? Perhaps "..(OLE) application as a potential tool..." reads better. Or, "...(OLE) as a potential ingredient...".
  • Line 77. Even a reference is given for the OLE obtention, briefly indicate the methodology and conditions to obtain the extract, which is substantial for the study
  • Line 102. Replace have been added by were added.
  • Line 104. Replace has been prepared by was prepared. Same for futher expressions.
  • Line 139. Any reference for colour measurements?
  • Line 313. Fibres in the extract? This is not right, check it.
  • Is being considered a sensory anaysis for further studies?
  • If possible, pictures of the developed samples may be included. These would be useful for the reader to match the colour results.

Author Response

Dear Reviewer,

thank you very much for your careful revision and helpful suggestions.

All the suggestions have been considered and accepted and the changes into the manuscript are highlighted in red. Attached the point-to-point response letter. 

Sincerely,

Carmine Summo

Reviewer 2 Report

The article presents a valuable study on the effect of the OLE addition to reduce nitrate/nitrite (NO) content on the physico-chemical features of the ripened pork sausages. The article was well planned and prepared. The authors correctly set the aim of the research and explained the innovativeness of the research. However, I do see some inaccuracies that should be corrected. I list the places where the article needs to be corrected.

Most importantly, why did the authors use a nitrate / nitrite supplement of 300 mg / kg in some samples? Is such an amount permitted to be used in accordance with Reg. (UE) n. 1129/2011?

The research methodology in several places was not well presented. Firstly, whether glucose or other sugar was used in the production of sausages to facilitate LAB growth?

Why was a 1 cm thick sample used to test the color parameters? Is this thickness sufficient?

I have doubts about the structure of the tables 4-7. Since the authors in the legend explain the adopted symbols for the samples, I think there is no need to repeat the contents of OLE and NO in each table.

The conclusion is not informative. It's written too broadly. As various reductions in the content of nitrates and nitrites have been applied with the simultaneous increasing level of OLE, it should be indicated which level is the most desirable.

Author Response

Dear Reviewer,

thank you very much for your careful revision and helpful suggestions.

All the suggestions have been considered and the changes into the manuscript are highlighted in red. Attached the point-to-point response letter. 

Sincerely,

Carmine Summo
